# Electrochemical Sensor Based on Molecularly Imprinted Polymer for the Detection of Cefalexin

**DOI:** 10.3390/bios9010031

**Published:** 2019-02-27

**Authors:** Bogdan Feier, Adrian Blidar, Alexandra Pusta, Paula Carciuc, Cecilia Cristea

**Affiliations:** Analytical Chemistry Department, Faculty of Pharmacy, Iuliu Hațieganu University of Medicine and Pharmacy, 4 Pasteur St., 400349 Cluj-Napoca, Romania; Feier.George@umfcluj.ro (B.F.); adiblidar@yahoo.com (A.B.); alexandrapusta@gmail.com (A.P.); paula_carciuc@yahoo.com (P.C.)

**Keywords:** cefalexin detection, molecularly-imprinted polymer, indole-3-acetic acid, boron-doped diamond electrode, glassy carbon electrode, cephalosporins, real samples

## Abstract

In this study, a new electrochemical sensor was developed for the detection of cefalexin (CFX), based on the use of a molecularly imprinted polymer (MIP) obtained by electro‒polymerization in an aqueous medium of indole-3-acetic acid (I3AA) on a glassy carbon electrode (GCE) and on boron-doped diamond electrode (BDDE). The two different electrodes were used in order to assess how their structural differences and the difference in the potential applied during electrogeneration of the MIP translate to the performances of the MIP sensor. The quantification of CFX was performed by using the electrochemical signal of a redox probe before and after the rebinding of the template. The modified electrode was characterized using atomic force microscopy (AFM), scanning electron microscopy (SEM), cyclic voltammetry (CV) and electrochemical impedance spectroscopy (EIS). The influence of different parameters on the fabrication of the sensor was tested, and the optimized method presented high selectivity and sensitivity. The MIP-based electrode presented a linear response for CFX concentration range of 10 to 1000 nM, and a limit of detection of 3.2 nM and 4.9 nM was obtained for the BDDE and the GCE, respectively. The activity of the sensor was successfully tested in the presence of some other cephalosporins and of other pharmaceutical compounds. The developed method was successfully applied to the detection of cefalexin from real environmental and pharmaceutical samples.

## 1. Introduction

Antibiotics have revolutionized the treatment of infectious diseases, but their overuse and misuse have resulted in the development of antibiotic resistance, one of the most significant public health problems nowadays, with over 25,000 annual deaths from infections caused by antibiotic-resistant bacteria in the European Union alone [1]. The extended use of antibiotics in a variety of fields, including human and veterinary medicine, as well as agriculture, has led to environmental contamination. Uncontrolled exposure to antibiotics found in the environment can cause major health issues such as allergies and resistance to broad-spectrum antibiotics. For this reason, the World Health Organization has recommended regulations regarding antibiotic use and surveillance [2]. In order to comply with these recommendations, it is crucial to develop fast and selective methods for antibiotic detection.

Cefalexin is an orally active β-lactam antibiotic, belonging to the first generation of cephalosporins and it is effective against Gram-positive bacteria, and, to a limited extent, Gram-negative bacteria. Its mechanism of action is the common mechanism of all β-lactam antibiotics: it blocks the synthesis of the bacterial cell wall [3]. It is frequently used in the treatment of upper respiratory tract infections, pneumonia and uncomplicated urinary tract infections [4].

In recent years, numerous methods for the quantification of cefalexin and other cephalosporins have been developed. These methods can be subdivided into different categories: microbiological, chromatographic, spectrophotometric, and electrochemical methods. Microbiological assay systems [5] have been employed for the detection of cephalosporins in milk, but presented as main disadvantages the lack of selectivity and the necessity for bacterial cell cultures. Instrumental techniques have also been used for the quantification of cephalosporins in different matrixes. These techniques include chromatographic methods such as high performance liquid chromatography (HPLC) with ultraviolet detection (UV) [6], HPLC coupled with mass spectrometry (MS/MS) [7,8] molecularly imprinted solid phase extraction (SPE) with HPLC [9] and UV spectrophotometry [10]. Unfortunately, all the aforementioned chromatographic techniques require complex work protocols, qualified staff and the use of harmful reagents in large quantities. The spectrophotometric method requires a previous derivatization of the sample with 1,2-naftoquinone-4-sulphonic.

Detection of cefalexin and other cephalosporins can also be performed using direct or indirect electrochemical methods [11,12,13]. These are an attractive option since they are relatively simple to use and offer rapid responses, while also having high sensitivity and reproducibility.

In a previous study [11], we reported a direct electrochemical method for cefalexin and other cephalosporin detection using a bare boron-doped diamond electrode. In this case, a decrease of the cefalexin peak in the presence of other cephalosporins was noticed, the selectivity of the method needing improvements. In order to achieve this goal, many methods can be taken into consideration, such as the use of biological agents like enzymes or antibodies, which are known for their high specificity. However, these compounds have certain drawbacks such as low stability and high cost. A promising alternative to the use of biological agents is the synthesis of molecularly imprinted polymers (MIPs), artificial receptors which can be designed to mimic natural selectivity. The MIP synthesis process requires the polymerization of a monomer in the presence of the analyte, which acts as the template. After the polymerization step, the template is removed, leaving cavities which will be complementary in shape, size and chemical functionality to the analyte [14]. The main advantages of MIP technology include high selectivity, resistance to conditions such as temperature and pressure and lower cost compared to biological agents [15].

The boron-doped diamond electrode (BDDE) has been used for the quantification of cephalosporins and penicillins [11,16], thanks to its unique characteristics such as low background and capacitive currents, higher resistance to fouling compared to other electrodes, wide potential window and high resistance to chemical and physical damaging agents [17].

In this study, an MIP-based electrochemical sensor was developed for the sensitive and selective detection of cefalexin (CFX). Indole-3-acetic acid (I3AA) was chosen as the monomer because it presents functional groups capable to interact with CFX (Appendix A) and it can be electropolymerized [18]. The MIP was obtained through “green chemistry”, the electropolymerization of the monomer (I3AA) being performed in aqueous solution, on a glassy carbon electrode (GCE) or on a boron-doped diamond electrode (BDDE) in the presence of CFX, as the template molecule. The two electrodes were used in order to assess how different electrode material and structure translate to the performance of the MIP sensor. Due to its wide electrochemical potential window, the BDDE allowed also to assess the influence on the MIP sensor of the high potential applied during the electrogeneration of the MIP and implicitly, of the overoxidation of the MIP film.

Each step in the modification of the electrode with MIP was characterized using atomic force microscopy (AFM), scanning electron microscopy (SEM), and electrochemical impedance spectroscopy (EIS) and each step was optimized. The quantification of CFX was performed by using the electrochemical signal of a solution of potassium ferricyanide before and after the rebinding of the template.

The developed sensor presents good selectivity, with low interferences from other antibiotics and pharmaceutical compounds and very good sensitivity, with a low limit of detection (LOD). The MIP-based sensor was successfully applied for the detection of cefalexin from pharmaceuticals and river water samples. To the best of our knowledge, this is the first time an MIP-based electrochemical sensor was developed for CFX detection.

## 2. Materials and Methods

### 2.1. Chemicals and Reagents

All reagents were of analytical grade and were used as received. All solutions were prepared with ultrapure water (18.2 MΩ, Millipore Simplicity). Cefalexin monohydrate, ceftriaxone sodium salt, cefadroxil monohydrate, cefotaxime sodium salt, cefaclor monohydrate, cefuroxime sodium salt, ceftazidime pentahydrate were provided by Antibiotice SA (Iași, Romania).

Ampicillin trihydrate, monosodium Hexaammineruthenium(III) chloride ([Ru(NH3)6]Cl3), 1,1′-Ferrocenedimethanol, HCl, Na_2_HPO_4_, NaH_2_PO_4_, NaCl, I3AA were purchased from Merck; K_4_[Fe(CN)_6_], K_3_[Fe(CN)_6_] and disodium phosphate, sodium dodecyl sulfate and methanol from Sigma-Aldrich (St. Louis, MO, USA), phosphoric acid, acetaminophen and ascorbic acid were purchased from Merck (Whitehouse Station, NJ, USA).

Capsules containing 500 mg cefalexin (Cefalexina Atb^®^ (Antibiotice SA)) and Someș River water collected near Cluj-Napoca, Romania were used for real samples analysis.

### 2.2. Apparatus and Electrochemical Measurements

The electrochemical experiments were performed with an AUTOLAB PGSTAT 302N (EcoChemie, Utrecht, The Netherlands) equipped with the NOVA 1.10 software. The BDDE (3 mm diameter, with approximately 0.1% boron content) was purchased from Windsor Scientific (Slough Berkshire, UK), the GCE from BAS Inc. (West Lafayette, IN, USA) and they were used as working electrodes in the conventional three-electrode cell, in static mode, along with Ag/AgCl KCl 3 M as reference electrode and a Pt wire as counter electrode, which were purchased from BAS Inc. (West Lafayette, IN, USA). Before each analysis, the BDDE and the GCE were polished using a 0.05 µm alumina suspension and polishing cloth, followed by an intense rinsing step using ultrapure water.

The electrochemical cell contained 5 mL of supporting electrolyte and for the characterization steps, the exact concentration of redox probe was added to the supporting electrolyte solution. For the polymerization procedure, the cell contained the same volume of supporting electrolyte solution containing the mentioned concentrations of monomer and template, for MIP fabrication, and only the mentioned concentration of monomer for the NIP fabrication. All electrochemical measurements were performed without the deaeration of the solution.

The main electrochemical techniques employed in this study were cyclic voltammetry (CV), differential pulse voltammetry (DPV) and electrochemical impedance spectroscopy (EIS). The CV technique was used for the polymerization procedure and also for the characterization of the unmodified and the modified electrode.

The CV parameters for the polymerization procedure were: a potential window between −1.6 to +1.6 V vs. Ag/AgCl, for the BDDE and between −1.0 V to +1.2 V vs. Ag/AgCl for the GCE, with a scan rate of 100 mV/s, for both electrodes. The number of cycles was optimized, varying between 2 and 10 cycles, with 5 cycles giving the best results.

For the redox probe selection, the CV parameters had to be changed according to the behavior of each probe. Thus, the potential window was between −0.6 V to +0.8 V vs. Ag/AgCl for the K_4_[Fe(CN)_6_]/K_3_[Fe(CN)_6_], between −0.9 V to +1.0 V vs. Ag/AgCl for the ([Ru(NH_3_)_6_]Cl_3_) probe and between −0.7 V to +1.0 V vs. Ag/AgCl for 1,1′-ferrocenedimethanol probe. The scanning rate was 100 mV/s and the number of cycles was two for all three probes tested. Also, these parameters were the same for the both types of electrodes, GCE and BDDE, and for all steps of the sensor development—unmodified electrode, after polymerization, after extraction and after incubation.

The EIS and DPV techniques, which were used after the selection of the redox probe, were employed for the electrode characterization and to assess the response of the unmodified electrode and of the modified one, after each step, DPV being also used for CFX quantification.

Regarding the electrochemical technique used for CFX quantification, linear sweep voltammetry and square wave voltammetry were also tested, but DPV led to the best results in terms of sensitivity and reproducibility. Similarly, EIS, which used in the optimization part, is a sensitive technique, very useful for surface characterization, but also with a longer duration and with more complex technical requirements, making it more difficult to employ it in routine and in situ analysis. Thus, for the quantitative analyses DPV was employed, representing the optimal procedure considering sensitivity, reproducibility, time of analysis and accessibility.

For the optimization of the DPV technique, the main parameters of the method, represented by the pulse height (PH), the pulse width (PW) and the scan rate (SR), were varied, concluding that following values were optimal: a potential window between +0.55 V to −0.30 V vs. Ag/AgCl, SR of 0.02 V s^−1^, PH of −0.100 V and PW of 50 ms. We chose to apply the DPV in reduction, in order to observe the reduction peak of the chosen redox probe, due to the lower probability of interferences in this approach.

EIS measurements were carried out in the presence of 10 mmol L^−1^ [Fe(CN)_6_]^3−/4−^ in PBS (0.02 mol L^−1^, pH 7.4) as redox probe, the impedance being measured in a frequency range of 0.01 and 100,000 Hz using the open circuit potential.

### 2.3. Elaboration of the MIP-Based Sensor

Before generating the MIP film, the working electrode (BDDE or GCE) was polished using a 0.05 µm alumina suspension and polishing cloth, followed by an intense rinsing step using ultrapure water. After the cleaning step, the MIP formation was achieved by immersing the working electrode in a 20 mM phosphate buffer solution (PBS), pH 7.2 containing the corresponding amount of the monomer (I3AA) and the template (CFX) and an electro-polymerization procedure was performed as previously reported [18]: CV in a potential window between −1.6 to +1.6 V vs. Ag/AgCl, for the BDDE and between −1.0 V to +1.2 V vs. Ag/AgCl for the GCE, with a scan rate of 100 mV/s, for both electrodes. Different concentrations of monomer and template and different ratios between them were tested. The concentrations of the monomer tested were 0.1, 1 and 5 mM. The concentrations of the template tested were 0.01, 0.05, 0.1, 0.5 and 1 mM. The resulting tested ratios were (monomer:template): 1:1, 2:1, 20:1 and 100:1. Also, included in this step, three different values for the number of cycles for the CV procedure were tested: 2, 5 and 10. The same procedure was used for the formation of the non-molecularly imprinted polymer (NIP), except the polymerization solution did not contain the template (CFX).

The removal of the CFX (template) from the MIP film was performed by keeping immersed the modified electrode in an exact volume of 2 mL of the corresponding solvent or solution (methanol, NaOH 0.1 M and PBS), in identical cells, under a constant stirring, at a speed of 7000 rpm, achieved with identical magnetic bars (stirrers), at room temperature for 5, 15, 30 and 60 min.

Each time the electrode was immersed in a different solution, meaning after each step of the sensor fabrication, polymerization, extraction, incubation and after each characterization procedure, the working surface was thoroughly rinsed with exact 10 mL of ultrapure water, using each time the same procedure. This was done in order to assure the complete removal of any unwanted residues from the previous step, to prevent the contamination of the solutions and to achieve a better reproducibility.

As working surfaces, the study focused on the use in parallel of two different working surfaces represented by two different types of electrodes, GCE and BDDE. All the mentioned steps and procedures were carried on using both of them, according to the mentioned conditions.

### 2.4. Surface Characterization Measurements

Atomic force microscopy (AFM) measurements were performed using an NTEGRA Spectra platform (NT-MDT, Russia), in intermittent contact mode, under ambient conditions, with 1 Hz scan frequency, and 512 × 512 scan points. We used standard silicon probes HQ-NSC15/Al BS (MikroMasch) with a typical force constant of 40 Nm^−1^, typical resonance frequency of the cantilever of 325 kHz, and tip radius less than 10 nm. AFM data processing was performed with the integrated software Nova v1.1.0.1837 (NT-MDT).

Scanning electron microscopy (SEM) analysis was performed with a UHR Scanning Electron Microscope model SU8230 (Hitachi, Japan). The samples were sputtered with gold (7 nm) before analysis.

### 2.5. Analysis Procedure

After the generation of the MIP film and extraction of the template, the electrode modified with MIP was immersed in PBS containing 10 mM [Fe(CN)_6_]^3−^ and DPV analysis was performed, recording the I_0_ current. After electrode rinsing with 10 mL of ultrapure water, the electrode was kept in 2 mL of solutions of different concentrations of CFX, prepared in ultrapure water or PBS, under a constant stirring, at a speed of 7000 rpm, at room temperature for 5, 15, 30 and 60 min. After that, the electrode was rinsed and it was immersed in PBS containing 10 mM [Fe(CN)_6_]^3−^ and DPV analysis was performed, recording the I_C_ current. In the end, the signal was calculated as (I_0_ − I_C_)*100/I_0_.

### 2.6. Selectivity Studies

The MIP-based sensor was prepared using CFX as template and the same detection procedure was used as the one described at 2.4, except the rebinding step was performed in 500 nM solutions containing ampicillin, cefadroxil, ceftriaxone, cefotaxime, cefaclor, cefuroxime, ceftazidime, gentamicin, vancomycin, ascorbic acid, acetaminophen.

### 2.7. Real Sample Analysis

For the analysis of the environmental samples, cefalexin was added to untreated river water to obtain a 5 mM solution. This solution was further diluted with the same untreated river water in order to obtain a 100 nM solution, which was analyzed using the developed MIP-sensor. To this solution standard solution of cephalexin in ultrapure water was added to obtain 200 nM, 300 nM, 400 nM spiked river water. The concentration of cefalexin was determined by the standard addition method (n = 3).

For the analysis of the pharmaceutical samples, the content of five capsules of Cefalexina Atb^®^ (Antibiotice SA) containing 500 mg/capsule were fine grounded and using the declared concentration, the accurately weighted quantity of powder corresponding to 5 mM cefalexin was dissolved in ultrapure water. After 40 min of sonication, 100 µL of the supernatant was diluted with ultrapure water to a volume of 5 mL, in order to obtain a concentration of 0.1 mM cephalexin. This was further diluted in order to obtain a final concentration of 100 nM, which was used for tests. Next, the same procedure was applied as in the case of river water.

## 3. Results and Discussion

### 3.1. Elaboration of the MIP-Based Sensor

#### 3.1.1. Electropolymerization of the I3AA

The modification of the GCE and of the BDDE with MIP film was performed by the electropolymerization of the I3AA in aqueous medium, using a procedure comprised of multiple CV cycles in a phosphate buffer solution (pH = 7.4) that contained the monomer and the template (Figure 1). The applied potential range was adapted to the electrode material: in the case of the GCE, the potential was scaned between −1.0 V and +1.2 V vs. Ag/AgCl, while in the case of the BDDE between −1.6 to +1.6 V vs. Ag/AgCl, due to the wide potential window and high mechanical and chemical stability of the BDDE. Similarly, in order to obtain the NIP film, the same procedure was applied, but in a phosphate buffer solution (pH = 7.4) that contained only the monomer.

In Figure 1, a well-defined oxidation peak is observed at around 0.8 V vs. Ag/AgCl on both electrode materials, corresponding to the anodic oxidation of the I3AA. This peak decreases from the second to the fifth scan, suggesting the formation at the surface of the electrode of an insulating film. In the case of the BDDE, a small peak is obtained in the first scan at around 1.5 V vs. Ag/AgCl, corresponding to the anodic oxidation of CFX, as previously reported [11]. Similar voltammograms were obtained in the case of electro-generating the NIP film (data not shown).

The proposed mechanism (Appendix A) for the formation of the MIP film involves the electro-oxidation of I3AA involving the loss of 2e^−^ and 1H^+^, with the oxidation product 3-methylenindolenine carboxylic acid, which further undergoes step-wise electropolymerization [18].

#### 3.1.2. Interaction Mechanism between the MIP Film and CFX

Taking into consideration the structure of CFX and of the MIP film, multiple interactions can be imagined between the template and the polymer: H bonds, electrostatic interactions, π-π stacking, all contributing to the selective rebinding of CFX (Figure 2).

### 3.2. Electrode Characterization

#### 3.2.1. Electrochemical Characterization

Electrochemical measurements were carried out in order to characterize the modified electrodes at different stages of the fabrication of the MIP-sensor. CV, DPV and EIS techniques were used to assess the response and to characterize the unmodified surfaces and after polymerization, extraction and incubation.

A first step in the electrochemical characterization was taken in the preliminary tests, in which CV tests were conducted in order to choose the most suitable redox probe. In this step, described in more detail, in the optimization section, three different probes were tested, [Fe(CN)_6_]^3−^/[Fe(CN)_6_]^4−^ yielding the best results. During these tests, all the probes tested showed a decrease in the peak current after polymerization, both for the MIP and NIP films, demonstrating the insulating effect of the polymer. Similarly, they all presented a major increase in the peak current after the extraction procedure, only for the MIP film, but due to their different properties (charge, size) and their different interactions with the polymer, the best response were obtained for the [Fe(CN)_6_]^3−^/[Fe(CN)_6_]^4−^ probe. This was once again observed, when after the incubation procedure, the peak current decreases, a decrease which was significantly bigger for the [Fe(CN)_6_]^3−^/[Fe(CN)_6_]^4−^ probe.

The results obtained using the CV technique and [Fe(CN)_6_]^3−^/[Fe(CN)_6_]^4−^ as redox probe, were confirmed by the DPV and EIS tests, using both BDDE and GCE. Using the DPV technique to measure the signal caused by the reduction of the [Fe(CN)_6_]^3−^, a high peak current was observed for the unmodified electrode (Figure 3a), which was severely decreased after polymerization (Figure 3b). After the extraction procedure, the intensity of the peak current increased by a significant degree (Figure 3c), but not reaching the intensity obtained for the unmodified surface. This difference between the signal for the unmodified electrode and for the MIP-modified electrode, after extraction, a difference which was also visible in the CV tests, indicates that the extraction procedure removed only the template molecules, decreasing the insulating properties of the MIP film, but not removing the film in its entirety, the extracted imprinted film retaining, although to a lesser degree, its insulating properties. After the incubation with CFX, a decrease in the peak current was once again observed; this can be explained by the fact that a large percentage of the imprinted cavities are reoccupied during the incubation process.

The EIS tests showed a similar behavior. In the Nyquist plots, the semicircle portion, at higher frequencies, corresponds to the electron transfer limited process, and the linear portion, at lower frequencies, may be ascribed to diffusion. Thus, the diameter of the semicircle equals to the electron transfer resistance (Rct), which is correlated with the dielectric and insulating features of the electrode/electrolyte interface, mainly, in our case, of the modifying film. For the unmodified electrode, a low value of Rct was obtained (Figure 4a), but after polymerization (Figure 4b), this value increased drastically, confirming once again the strong insulating properties of the polymeric film. After extraction, a decrease of the Rct was observed (Figure 4c), but the response did not reached the values of the unmodified electrode, showing once again the persistence of the imprinted film on the working surface. The capacity of the MIP to rebind the CFX molecule was proven by the increase of the Rct after the incubation step (Figure 4d).

All these results demonstrate the modification of the electrode with a MIP film, the successful extraction of the template and the capacity of the MIP to recapture the analyte molecules, CFX. They also show that by removing the template and making available the imprinted cavities, the polymeric film increases its porosity and decreases its insulating properties, the unoccupied cavities allowing an easier electron transfer between the redox probe and the electrode and these changes can be quantified using the signal of the redox probe.

#### 3.2.2. Surface Characterization

SEM images at higher magnification (100 k×) were used to get an insight into the surface morphology of NIP (Figure 5C) and MIP (Figure 5B) films and, more interesting, into the surface of the MIP film after the extraction procedure (Figure 5C).

NIP film surface looks compact and smooth. The MIP film, at a first glance seems similar to the NIP surface, but a more porous morphology and some roughness of the surface can be observed. However, the MIP surface after extraction, definitely presents a higher degree of roughness and a further increase in porosity, indicating that the extraction procedures of the template molecules leads to a change in the morphology of the MIP film.

In contrast to SEM, which provides a two-dimensional projection or a two-dimensional image of a surface, the main advantage of AFM is the fact that it provides a true three-dimensional profile of the surface. Through AFM is it possible to carry out topographic contrast direct height measurements, measure the thickness and roughness of the layer.

In the present work, the formation of both NIP and MIP layers was suggested by the AFM images (Figure 5D,E), indicating an elevated layer with the surface height of 3.1 nm for NIP and 15 nm for MIP. The root mean square (RMS) roughness obtained was of 0.33 nm for NIP and 0.65 nm for MIP.

These results can be easily explained by the inclusion of the template molecule in the MIP film, drastically modifying the polymerization process and implicitly the structure of the film. CFX, the template, has a more voluminous molecule than the monomer and also it does not make covalent bonds with the polymer, causing a considerably less structured organization of the polymeric film, causing an increase in both roughness and thickness, for the MIP film.

Similarly to the results obtained through SEM, the thickness of the MIP layer after extraction (Figure 5F) was 7 nm and the roughness 0.95 nm. These data support the success of the extraction procedure, which causes, by removing the template, the creation of the “imprinted” cavities, which further leads to an increase in roughness. This removal of the template, combined perhaps with the removal of certain residues remnant from the polymerization procedure and with a certain degree of reorganization of the polymeric film, after template removal, is the cause of the decrease in thickness of the MIP layer after extraction.

### 3.3. Optimization of the MIP Sensor

The development of the MIP-based sensor was obtained in several steps and each one needed to be optimized. The overall development of the sensor can be divided in four main steps: first, the selection of the redox probe for the indirect detection, second, the creation of the initial imprinted polymeric film, containing the template molecule in it, third, the extraction procedure (the removal of the template) and fourth, the incubation with CFX (the recapture or the rebinding of the template). In the optimization of each parameter, the final signal (the signal difference after the extraction and the incubation step) was taken into account, because each one of the four aforementioned consecutive steps influences the final response of the sensor.

The final signal was constructed as a ratio between the difference of the two signals, after extraction and after incubation, and the signal after extraction. Considering that the signal after extraction is almost constant, the denominator part of the ratio remains also constant and the differences in the numerator appear only as a result of the changes in the values of the signal after incubation.

To better characterize the formation of imprinted cavities and to demonstrate that in the incubation process there is truly taking place a specific adsorbtion phenomenon, all the optimization tests carried out for the MIP-electrodes, were also realized using NIP-electrodes. Working in parallel with an MIP and NIP, we could also more easily elucidate the influence of each parameter on the performance of the MIP.

Because in DPV, the technique used for quantification, the signal after extraction is always going to be greater in absolute value and for simplicity we chose the difference to be calculated as the signal after extraction minus the signal after incubation. Because the signal after incubation decreases with increasing concentrations, the difference and also the ratio, increase their values in correlation with the concentration of analyte. In a similar way, the signal was constructed for the CV technique.

In contrast, because in EIS measurements Rct increased in value due to increasing amounts of captured analyte, the difference was reversed in the construction of the signal in the EIS technique. For the optimization part, in the same way, two other signals in each technique were constructed, corresponding to each procedure: a signal after polymerization, using the signal after polymerization and the signal of the unmodified electrode and a signal after extraction, using the signal after extraction and after polymerization. The formulas for the calculation of the signals are summarized below.

For DPV and CV techniques:(1)$$S_{\mathrm{incub}}=\frac{{\;I}_{\mathrm{extr}}-I_{\mathrm{incub}}}{I_{\mathrm{extr}}};\;S_{\mathrm{extr}}=\frac{{\;I}_{\mathrm{extr}}-I_{\mathrm{polym}}}{I_{\mathrm{polym}}};\;S_{\mathrm{polym}}=\frac{{\;I}_{\mathrm{unmod}}-I_{\mathrm{polym}}}{I_{\mathrm{unmod}}}$$

I_unmod_ is the peak height in DPV/CV for the unmodified electrode, I_polym_ is the peak height in DPV/CV after polymerization, I_extr_ is the peak height in DPV after extraction, I_incub_ is the peak height in DPV after incubation.

For EIS technique:(2)$$S_{\mathrm{incub}}=\frac{{\;R}_{\mathrm{incub}}-R_{\mathrm{extr}}}{R_{\mathrm{extr}}};\;S_{\mathrm{extr}}=\frac{{\;R}_{\mathrm{polym}}-R_{\mathrm{extr}}}{R_{\mathrm{polym}}};\;S_{\mathrm{polym}}=\frac{{\;R}_{\mathrm{polym}}-R_{\mathrm{unmod}}}{R_{\mathrm{unmod}}}$$

R_unmodif_ is the Rct in EIS on the unmodified electrode, R_polym_ is the Rct in EIS after polymerization, R_extr_ is the Rct in EIS obtained after extraction, R_incub_ is the Rct in EIS obtained after incubation. The response after polymerization is directly correlated to the insulating properties of the imprinted polymer formed after electropolymerization, a higher response value meaning a more insulating film, which is correlated to the thickness, composition and structure of the film. The response after the extraction is directly correlated to the amount of template removed during the extraction step, a higher response value being equal to a higher amount of template removed. In addition, most importantly, the final response is directly correlated to the amount of recaptured analyte, a higher response equals a higher amount of analyte recaptured by the MIP, which can be correlated to a higher concentration in the sample.

#### 3.3.1. The Selection of the Redox Probe

For the quantification and characterization of the sensor, three different redox probes were tested: [Fe(CN)_6_]^3−/4−^, an anionic redox probe, 1,1′-ferrocenedimethanol, a neutral redox probe and [Ru(NH_3_)_6_]^3+^, a cationic redox probe, to investigate the most suitable one for the characterization of the modifications suffered by the MIP film and for the CFX quantification. They were chosen based on their particular properties, which could cause different interactions, electrostatic attraction or repulsion, between the probe and the outer layer of the MIP film. Also, another important property of the redox probes is the distance at which their electron transfer is hindered, this property offering a possibility to estimate the thickness of the MIP layer.

As it can be seen from the data presented in Appendix A, all three redox probes provided a good signal after polymerization, meaning that regarding the polymerization layer, any of the three probes could be used.

The results for the NIP were similar, to those obtained for MIP, after polymerization. More precisely, a signal of 0.995 was obtained for both MIP and NIP, using [Fe(CN)_6_]^3−/4−^, on the GCE. Similar, using the same probe, a signal of 0.986 for the MIP and a signal of 0.993 for the NIP, were obtained on the BDDE. This is in accordance with the fact that both the polymer and the template molecule showed insulating properties, regardless of the probe used.

Nonetheless, after both the extraction and incubation procedures, only the [Fe(CN)_6_]^3−/4−^ probe can still detect the film and more importantly, can detect the rebinding of the template. This can be easily explained by the differences in the distances at which each redox probe can realize its electron transfer, Ru^3+^ and ferocene being able to conduct the direct electron transfer at a greater distance than the [Fe(CN)_6_]^3−/4−^ [19,20]. This finding is in accordance with the data from SEM and AFM, which estimates the thickness of the MIP film, before extraction at approximately 15 nm and under 10 nm after extraction. Another possible factor is the influence of the charge of the redox probe. The MIP film, due to its acidic moieties, causes electrostatic repulsion with the anionic probe ([Fe(CN)_6_]^3−/4−^), but does not interact electrostatically with the neutral probe (1,1′-ferrocenedimethanol) or will even attract the cationic probe (Ru^3+^). All these results led us to choose [Fe(CN)6]^3−/4−^ as redox probe for the next studies.

For the NIP modified electrodes, the behavior of the three probes tested was similar to that of the MIP, only the [Fe(CN)_6_]^3−/4−^ probe being able to detect the film, after the extraction step. However, very important, there were clear differences regarding the signal after extraction and after incubation, using the [Fe(CN)_6_]^3−/4−^ probe, for the NIP, comparatively to the MIP. First, a much smaller signal, was obtained after extraction for NIP, comparatively to the one obtained for the MIP, For the GCE, 6.320 for MIP and 1.138 for NIP were obtained and, similarly, on the BDDE, 8.348 for MIP and 0.735 for NIP. These results clearly show, that even though the signal of the NIP after extraction is changed, its modification is much smaller than that of the MIP, demonstrating the resulting imprinting process due to the removal of the template. Also, these results predicted a higher sensitivity, caused by a more pronounced imprinting process on the BDDE, in comparison to the GCE.

Second, after incubation, there were also clear differences, even though modifications in the signal were taking place on both of the electrodes. The values obtained, after incubation, for MIP and NIP, were 0.436 and 0.176, on the GCE, respectively, 0.532 and 0.123, on the BDDE. For this step, the modification can be explained by the changes of the polymer and its surface, caused by the incubation procedure, in a different solvent, but also by the phenomenon of nonspecific adsorption of the analyte, on the outer surface of the polymeric film. These two processes take place on both types of surfaces, MIP and NIP, the process that distinguishes them being the specific absorption, the template recapture inside the polymeric film. This way, the final signal for the NIP is caused by the nonspecific adsorption of the template and by the influence of the incubation, whereas the signal of MIP is caused mainly by the specific recapture of the template inside the polymeric film.

The behavior of the response of the NIP, in relation to the response of the MIP, was similar to that obtained in the first step, the one regarding the redox probe, in all the tests carried out in the other three steps of optimization.

#### 3.3.2. The Optimization of the Formation of the Imprinted Polymeric Film

The second optimized step in the development of the MIP sensor was the electropolymerization procedure. This is probably the most important step, because during this step a strong interaction between the template molecule, the monomer and the electrogenerated polymer must be assured. The most important parameters for this step are those regarding the composition of the polymerization mixture. The concentration of the monomer, the concentration of the template and implicit the ratio between these two components have been optimized in order to assure a good monomer-template interaction. Also, these parameters influence the thickness and the structure of the imprinted polymeric film resulting after polymerization. A thicker layer makes possible the creation of more imprinted “pockets”, but also can affect the accessibility to this “binding pockets” in the extraction and rebinding steps.

Another parameter that can be easily manipulated, in the case of electropolymerization through the use of cyclic voltammetry is the number of cycles of the procedure, which influences mainly the thickness of the layer and also its stability. The optimization of the main parameters regarding the polymerization procedure is presented in Appendix A.

As it can be seen from the data presented in Appendix A, the first optimized parameter was the concentration of the monomer, with three concentrations being tested (0.1, 1 and 5 mM), while the concentration of CFX was kept constant at 0.05 mM, varying in this way also the ratio monomer:template. Using the same concentrations of the monomer but no template, three NIP-sensors were fabricated, tested and compared. For comparison all three signals – after polymerization, after extraction and after incubation were taken into account. As it can be seen the value of S_polym_ was very high in all cases, especially due to the highly insulating nature of the polymer. Also, as it can be seen from the similar S_polym_, for different number of cycles, the insulating properties of our polymer were less influenced by the thickness of the layer, a high response being obtained even only after 2 cycles.

Considering S_extr_ and S_incub_, a number of 5 cycles proved to give better results compared to 2 or 10 cycles. This can be explained by the equilibrium that occurs between the number of cycles and the amount of template that can be removed and thus, also rebounded. A lower number of cycles lead to a less structured polymer, which means not enough cavities formed or a large number of not fully formed cavities, which are not stable enough during the extraction step or simply are not able to rebind the template, due to a smaller number of binding sites. On the other hand, a higher number of cycles can lead to a higher value of thickness and to an excessively structured polymer, which due to its high thickness and high rigidity, hinders the proper extraction and recapture of the template molecule.

Also, regarding the S_incub_, which varies the most and is the most important response, being the final criteria in choosing the optimal parameters, good polymerization and extraction do not necessarily assure a good rebinding of the analyte. From the data obtained, a concentration of 1 mM for the monomer and of 0.5 mM for the template, with a procedure using five cycles, provided the best results in terms of the amount of recaptured analyte.

#### 3.3.3. The Optimization of the Extraction Procedure

The solvent and the duration of the incubation were the main parameters of the incubation procedure that were optimized (Appendix A). Three options were tested as solvents for extraction, based on the solubility of CFX, CFX being highly soluble in MeOH and alkaline aqueous solutions. The best results were obtained using an aqueous solution of NaOH 0.1 M. This can be explained by the fact that the use of a pure non-aqueous solvent such as MeOH affects severely the integrity of the imprinted film, which even though it can cause a better response after extraction, it also affects the binding capacities of the imprinted cavities. Instead, the use of an alkaline aqueous solution provided the balance between stability, being a similar medium with the one in which the polymerization was realized, and efficiency, the change in pH, towards more basic, offering a strong enough change in the solubility of the template in order to determine the break of the bonds between the polymer and template.

Regarding the duration of the procedure, 30 min offered enough time for a satisfactory extraction, without any significant damage to the film. This can be seen by the fact that up to 30 min the amount of rebound template increases, but for 60 min, the amount is lower, which may indicate that the prolonged period of immersion in the strong alkaline solution may have affected the integrity of the film.

#### 3.3.4. The Optimization of the Incubation Procedure

For the incubation procedure, similar parameters as for the extraction were optimized. The solvent tested was PBS, the same as the one used for polymerization and ultrapure water. Even though PBS yielded slightly better results, water offered the advantage of bearing more resemblance to the matrix of the targeted samples. That is why, taking into account the applicability of the method, water was chosen as incubation medium. Regarding the duration of the incubation, the captured amount of analyte seemed to reach a plateau at 30 min, there being no large differences between the values at 30 and 60 min. Therefore, the final parameters chosen for the incubation procedure were water as the medium and 30 min of incubation (Appendix A).

### 3.4. Analytical Performance of the MIP-Based Sensor

The analytical performance of the developed sensor was tested under the optimized conditions.

#### 3.4.1. Calibration Curve and Limit of Detection

Because the process of rebinding the template can be simulated as an adsorption process, for the MIP surfaces it is required to establish the adsorption isotherm governing the rebinding of the template. The data obtained for our work presented a logarithmic growth, the Freundlich adsorption isotherm, being the one that fitted the best our results. The fact that this empirical equation fitted our data can be explained by the heterogeneity of our imprinted surface, in which characteristics of the Langmuir model, such as a monolayer of adsorbate and equivalency between adsorption sites cannot be fulfilled, the Freundlich model being developed for more irregular and complex surfaces. The unfitted data presented a logarithmic growth of the signal in relation to the concentration of the template, reaching a plateau at higher concentrations, a sign of the fact that all active imprinted cavities have been filled. This type of behavior is characteristic to MIP-based sensors [21,22,23].

That is why, for the calibrations curves, in order to obtain a linear correlation, the signal was plotted against the negative value of the decimal logarithm of the concentrations, four different calibration curves being constructed, one for each type of modified surface, in the same range of concentrations: 10–1000 nM.

The linear relationships found, with their correlated equations and their correlation coefficients (R2), were as it follows: for MIP-BDDE, S_incub_ = −0.1809 × (−log(C) (M)) + 1.7392; with R^2^ = 0.90669, for NIP-BDDE, S_incub_ = −0.05566 × (−log(C) (M)) + 0.62457; with R^2^ = 0.93433, for MIP-GCE, S_incub_ =−0.1483 × (−log(C) (M)) + 1.44859; with R^2^ = 0.97922 and for NIP-GCE, S_incub_ = −0.13175 × (−log(C) (M)) + 1.16155; with R^2^ = 0.99273 (Figure 6).

For the determination of the limit of detection (LOD) the following equation was used: LOD = 3S_b_/m, in which *S_b_* is the standard deviation of the response of the blank solution and *m* is the slope of the calibration curve. An LOD of 3.2 nM and 4.9 nM was obtained for the BDDE and the GCE, respectively. These results are better than the one obtained by other electrochemical methods and even instrumental methods, as seen in Appendix A.

From the results presented it can be deduced that the BDDE-modified surface presented better sensitivity, even if you take into account only the MIP signal or the difference in response for the MIP and NIP surfaces. This can be attributed to larger number of imprinted cavities, for the film on the BDDE surface, caused by several factors, such as the larger potential window, the better conducting properties of the BDDE and also to the possibility of a higher incorporation of the template, in the process of its electroxidation. However, the GCE offered a greater reproducibility and a better correlation for the results, which can also be easily explained by the more homogeneous composition of the GCE surface, compared to the BDDE surface, known for its more heterogeneous nature, which causes some researchers to avoid, it when wanting to develop a biomimetic sensor. Considering, all this, it can be said that even though the BDDE offered a slightly greater sensitivity, one should choose the GCE, when wanting to develop a biomimetic platform, because of the very heterogeneous composition of the BDDE surface and slightly less reproducible response, which could affect the performance of the developed analytical platform.

#### 3.4.2. Precision and Reusability of the Imprinted Sensor

In order to check the reproducibility of the developed method, 5 different modified BDDEs and 5 different modified GCEs were fabricated and applied to the analysis of a 100 nM CFX solution. The relative standard deviations for the two electrodes were 4.92% and 2.66%, respectively. This shows that even though the LOD obtained with the BDDE is lower, its precision is worse compared to the GCE.

In order to test the reusability of a MIP-based sensor, after a first CFX analysis, an extraction procedure was applied to the sensor and a second CFX analysis was performed. The results presented a great variability, with no reliable results for the second analysis using the same sensor.

#### 3.4.3. Selectivity Studies

The selectivity of the MIP-based sensor was tested by performing the incubation step in solutions containing other cephalosporins antibiotics (cefadroxil (CFD), cefaclor (CFC), ceftriaxone (CFTXO), cefotaxime (CFTX), ceftazidime (CFTZ), cefuroxime (CFOX)), ampicillin (AMP) that belongs to the penicillin class, two other antibiotics from a different classes (GEN, VAN) and two pharmaceutical compounds intensively used (ascorbic acid (AA) and acetaminophen (APAP)).

The incubation with CFTXO, CFTX, CFTZ or CFOX led to no significant signal modification, because even though these molecules present the common cephalosporin nucleus, they have different substituents, impairing them to be bound in the cavities of the MIP film (Figure 7). In contrast, CFD and CFC, two cephalosporins with a similar and respective identical one side chain to the one of CFX, were bound in the cavities of the MIP, suggesting the importance of the side chain in the fabrication of the MIP film. This is supported by the results obtained after incubation with AMP, a penicillin with the side chain identical to the one of CFX and different core nucleus, AMP being bound also by the MIP film.

In the case of the compounds that are not structurally related to CFX, they presented low affinity for the MIP film fabricated on both electrodes, the sensor proving to be selective for CFX.

### 3.5. Real Sample Analysis

The developed MIP-sensor was tested by analyzing two main types of real samples, containing CFX, environmental samples, represented by river water (Someș River, Cluj-Napoca, Romania) and pharmaceutical formulations, represented by capsules (Antibiotice Iași, Romania, 500 mg). The concentration of CFX in the tested samples was determined using the standard addition method. Good recoveries were obtained for both types of samples (Table 1) after a minimal samples treatment, this features being a huge advantage for the presented method.

## 4. Conclusions

A selective MIP using I3AA as a monomer was successfully electrogenerated on GCE and BDDE, with the latter leading to better sensitivity. The MIP film was characterized by surface and electrochemical analyses and it was optimized.

The developed sensor proved to be selective towards CFX in the presence of the other cephalosporin molecules and of other common interferents. The developed method presents a low limit of detection and was successfully applied to detection of CFX from real environmental and pharmaceutical samples.

## Figures and Tables

**Figure 1 biosensors-09-00031-f001:**
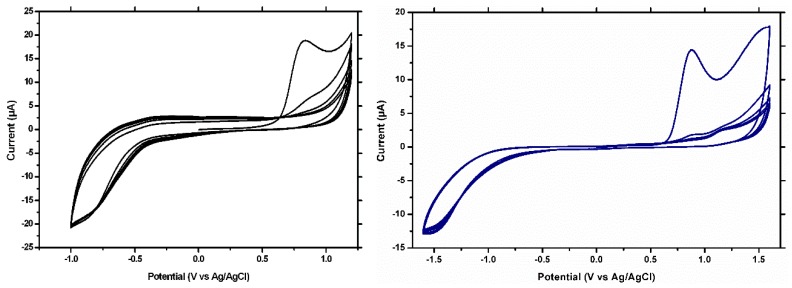
CVs for MIP formation on the GCE (**left**) and on the BDDE (**right**).

**Figure 2 biosensors-09-00031-f002:**
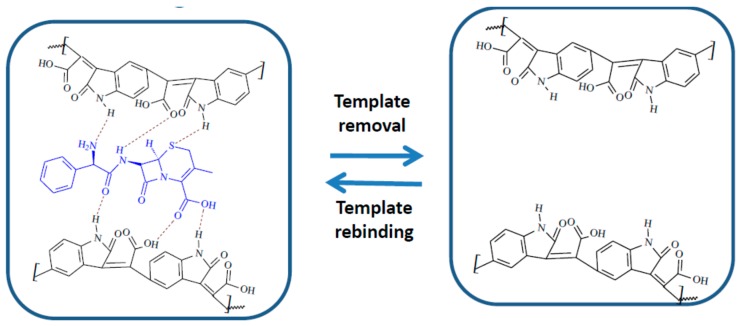
Interaction mechanism between the MIP film and CFX.

**Figure 3 biosensors-09-00031-f003:**
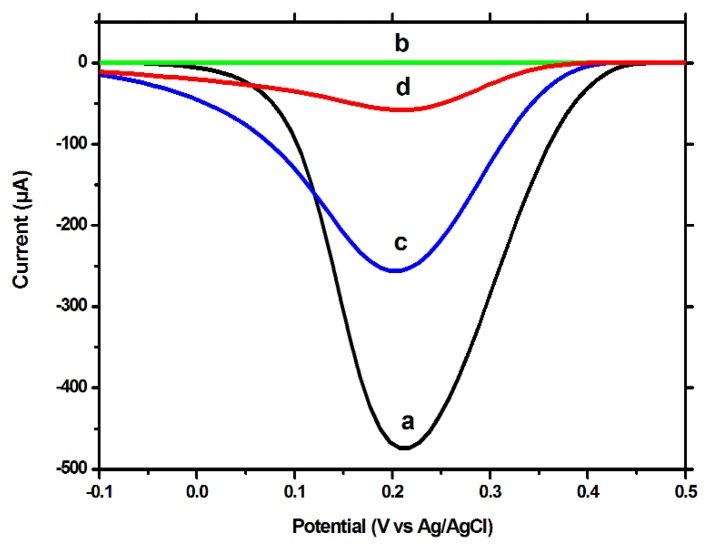
DPV voltammograms on the BDDE in 10 mM [Fe(CN)_6_]^3−^, in 0.02 M PBS on: (**a**) the unmodified electrode, (**b**) after electropolymerization, (**c**) after template removal (incubation for 30 min, under stirring, in NaOH 0.1 M solution) and (**d**) after rebinding of CFX (incubation for 30 min, under stirring, in 1 µM CFX solution). PH −100 mV, PW 50 ms, SR 20 mV/s.

**Figure 4 biosensors-09-00031-f004:**
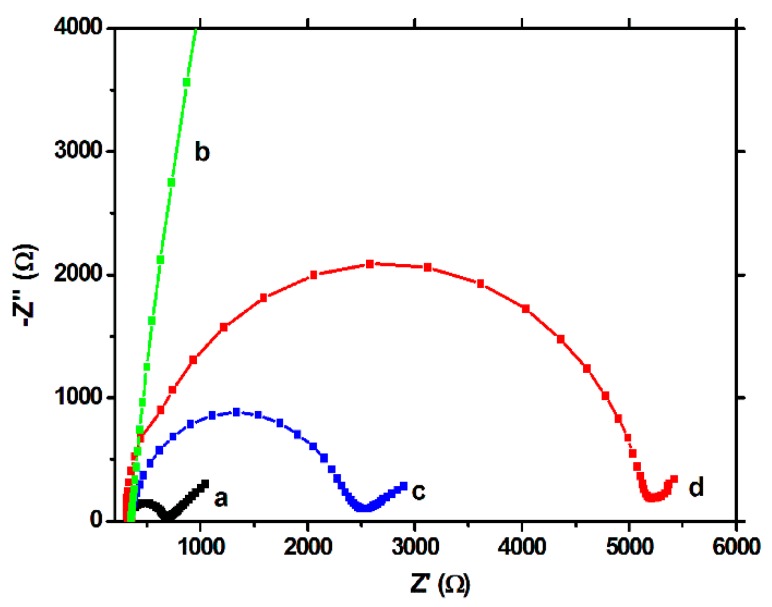
EIS spectra (50 frequencies) on the BDDE in 10 mM [Fe(CN)_6_]^3−^/[Fe(_CN)6_]^4−^ in 0.02 M PBS, on: (**a**) the unmodified electrode, (**b**) after electropolymerization, (**c**) after template removal (incubation for 30 min, under stirring, in NaOH 0.1 M solution) and (**d**) after rebinding of CFX (incubation for 30 min, under stirring, in 1 µM CFX solution).

**Figure 5 biosensors-09-00031-f005:**
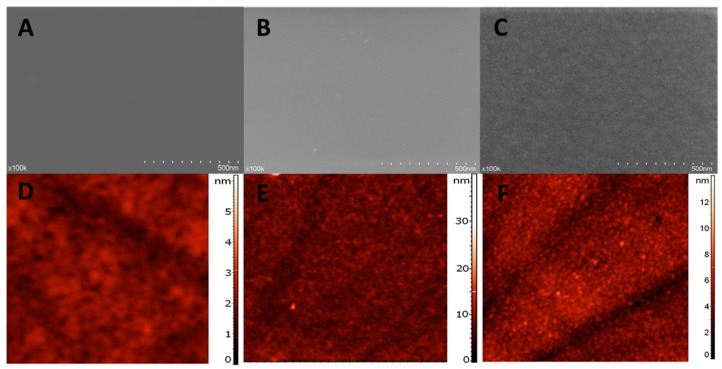
Surface characterization–SEM images: (**A**) NIP, (**B**) MIP and (**C**) MIP after extraction and AFM images: (**D**) NIP, (**E**) MIP and (**F**) MIP after extraction.

**Figure 6 biosensors-09-00031-f006:**
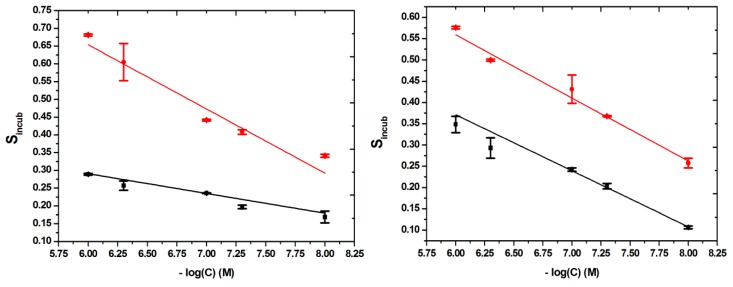
Linear relationship between MIP-based BDDE (**left**) and GCE (**right**) response (MIP (red) and NIP (black) and CFX concentration from 10 to 1000 nM).

**Figure 7 biosensors-09-00031-f007:**
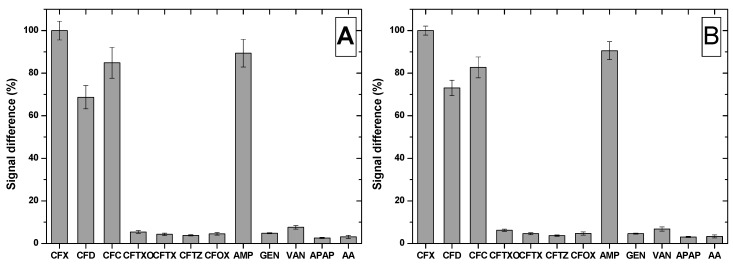
Selectivity tests using the MIP modified (**A**) BDDE and (**B**) GCE.

**Table 1 biosensors-09-00031-t001:** Real sample analysis.

Sample	BDDE			GCE		
	Added (nM)	Found (nM)	Recovery (%)	Added (nM)	Found (nM)	Recovery (%)
River water	100	107.14	107.14	100	106.57	106.57
Capsules	100	108.43	108.43	100	96.43	96.43

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
