# Peer review of "Electrochemical Sensor Based on Molecularly Imprinted Polymer for the Detection of Cefalexin"

_biosensors, 2019, doi:10.3390/bios9010031_

Round 1
Reviewer 1 Report
The manuscript describes an electrochemical sensor for detection of cefalexin based on electropolymerization of I3AA on two different electrodes, GCE and BDDE. It was reported a complete optimization of polymerization and rebinding conditions, as well as response, selectivity and application of the sensor. The work is interesting, but some points must be improved before the publication:
- There are few typos and grammatical errors throughout the manuscript. For example, milliliters are always written as "ml" instead of "mL".
- The manuscript is written quite colloquially, so it should be reviewed by a specialist in scientific texts who is native speakers of the English language.
- Keep a standard for concentration units, use mM, nM or 10-4, 10-7.
- The Fig. S2 was not cited in the manuscript (my suggestion is cited it on line 238).
- Fix the binding angles of Fig. 2 and Fig. S2. They are completely wrong.
- According to the manuscript, I expected to see the interactions between the template and the functional monomer. So, I suggest that the authors improve the text in the manuscript or include such interactions. However, the author has already shown the interaction in Fig. 2.
- Is there any incubation time to provide the interaction between the template and the functional monomer before the electropolymerization? Some papers about MIP by electropolymerization show that the sensor performance improves if both compounds are in contact for certain time to interact before the polymerization.
- What is the reason for the differential pulse voltammetry to have better results than square wave voltammetry if reversible compounds were used as a probe? A better sensitivity is expected for SWV when reversible compounds are used.
- What are the properties and interactions mentioned on line 260?
- It is a little weird to be able to observe the pores after the extraction on the MIP-electrode surface, because the pores should have dimensions of angstrom, since that is the size of the template. Usually, SEM does not reach these dimensions. What is the morphology of the NIP after “extraction” process?
- The NIP should be further explored throughout the manuscript. There is practically no discussion to prove the efficiency of cavity formation on the MIP. I missed this discussion mainly in the tables of the supplementary material in the optimization stage. When comparing the MIP and NIP values, it is possible to observe well the choice of optimal parameters.
- What is the concentration of CFX used for incubation after extraction reported in tables S3 and S4? By the way, the discussion on these results should be improve. It is not very clear.
- It is not clear the reason of 5 cycles is better than 10 cycles.
- Some sentences are too long, which makes the text confusing. For example, the sentence between lines 439 and 443. Avoid sentence too long.
- Some questions about the calibration curves presented in Fig. 6: (1) What are the black and red lines? I guess one is MIP and the other is NIP, but it is not clear in the caption. (2) How many time the experiment was made? As the standard deviation was added, the number of experiment should be included in the caption (n = 3?). (3) Why did NIP present a linear response too? Usually, NIP synthetized by electropolymerization does not present any response in the presence of analyte, since only the polymer is formed on the electrode surface, without any cavity. Thus, the NIP profile should be the same as that obtained after MIP polymerization (before the template extraction). Because of this, it is very important to include the same results that the authors showed for MIP in Fig. 3 for the NIP as well. (I suggest that the authors check how this result for the NIP was presented in the paper Sensors and Actuators B 253 (2017) 180–186). (4) I am surprised by the wide linear range of the sensor. Sensors based on electropolymerized MIP usually has a small linear range, because the cavity is easily occupied by the analyte, resulting in saturation of the electrochemical signal. How do the authors explain this behavior? (5) With the data of the calibration curve, it is possible to calculate the affinity constant of the MIP, which can be enriching for the work.
- It is not clear why I3AA was chosen as a monomer, since it does not form a very stable polymeric film on the electrode surface. Even with good results, what was the reason for choosing this monomer? The use of graphene or carbon nanotubes could have aided in the stability and fixation of the film on the electrode.
- Include the structure of all compounds used in selectivity analysis in supplementary information to facilitate the comparison and understanding among the results.
- How do the authors explain AMP responded to the sensor while penicillin, which as a very similar structure to AMP, did not present a signal?
- Why was the logarithmic scale concentration used in the recovery analysis (Table 1)? I could understand it in the calibration curves (since concentrations with different order of magnitude were used). However, there is no sense used logarithmic concentration in recovery analysis.
Based on my comments, I recommend the manuscript to be published after major review.
Author Response
The manuscript describes an electrochemical sensor for detection of cefalexin based on electropolymerization of I3AA on two different electrodes, GCE and BDDE. It was reported a complete optimization of polymerization and rebinding conditions, as well as response, selectivity and application of the sensor. The work is interesting, but some points must be improved before the publication:
We would like to thank to the reviewer for recognizing the quality of our study and for the useful suggestions which aim to improve the technical aspect of the work.
1. There are few typos and grammatical errors throughout the manuscript. For example, milliliters are always written as "ml" instead of "mL".
The manuscript was modified according to the reviewer’s suggestion.
2. The manuscript is written quite colloquially, so it should be reviewed by a specialist in scientific texts who is native speakers of the English language.
The entire manuscript was revised and the language level was improved.
3. Keep a standard for concentration units, use mM, nM or 10-4, 10-7.
The manuscript was modified according to the reviewer’s suggestion.
4. The Fig. S2 was not cited in the manuscript (my suggestion is cited it on line 238).
We thank the reviewer for pointing this to us. A citation for figure S2 was introduced.
5. Fix the binding angles of Fig. 2 and Fig. S2. They are completely wrong.
Figure 2 and figure S2 were drawn to show the possible interactions between the template molecule and the polymer. We did not use molecular modeling software, so these figures should be looked as schemes, with emphasis on the hydrogen bonds between different functional groups of the template and the polymer and not as an actual 3-D configuration of the polymer.
6. According to the manuscript, I expected to see the interactions between the template and the functional monomer. So, I suggest that the authors improve the text in the manuscript or include such interactions. However, the author has already shown the interaction in Fig. 2.
In order to avoid the redundancy in the manuscript there were mentioned only the interactions of the template with the polymer, which has the same functional groups as the monomer and therefore, the interactions are supposed to be similar.
7. Is there any incubation time to provide the interaction between the template and the functional monomer before the electropolymerization? Some papers about MIP by electropolymerization show that the sensor performance improves if both compounds are in contact for certain time to interact before the polymerization.
The role of the incubation period before the electropolymerization was studied in the preliminary tests, when the response of the sensor was tested after different periods of incubation (from no incubation period to even 48h). The results showed that there is no influence of the incubation time on the properties of the MIP. So, usually the mixture containing the template and the functional monomer was prepared just before polymerization and kept around 5 min before performing the electropolymerization.
8. What is the reason for the differential pulse voltammetry to have better results than square wave voltammetry if reversible compounds were used as a probe? A better sensitivity is expected for SWV when reversible compounds are used.
Even though SWV is recommended for reversible processes, it is less recommended for the analysis of slower electrode processes, which is our case due to the nature of the MIP film on the electrode. In the preliminary tests, we compared different electrochemical techniques and the results showed a better signal (especially in terms of reproducibility) when DPV was used.
9. What are the properties and interactions mentioned on line 260?
The text was modified with the properties of charge and size. As mentioned in the text, this point was “described in more detail, in the optimization section”.
10. It is a little weird to be able to observe the pores after the extraction on the MIP-electrode surface, because the pores should have dimensions of angstrom, since that is the size of the template. Usually, SEM does not reach these dimensions. What is the morphology of the NIP after “extraction” process?
It is not mentioned the claim that the pores observed correspond to the cavities imprinted after the template removal. The text was re-written for more clarity.
11. The NIP should be further explored throughout the manuscript. There is practically no discussion to prove the efficiency of cavity formation on the MIP. I missed this discussion mainly in the tables of the supplementary material in the optimization stage. When comparing the MIP and NIP values, it is possible to observe well the choice of optimal parameters.
The NIP is discussed in the chapters presenting the preparation, the optimization, calibration curve. The discussion to prove the efficiency of cavity formation on the MIP was improved.
12. What is the concentration of CFX used for incubation after extraction reported in tables S3 and S4? By the way, the discussion on these results should be improved. It is not very clear.
The concentration of CFX used for incubation is 0.5 µM.
13. It is not clear the reason of 5 cycles is better than 10 cycles.
The text was modified for more clarity.
14. Some sentences are too long, which makes the text confusing. For example, the sentence between lines 439 and 443. Avoid sentence too long.
The entire manuscript was revised and modified for more clarity.
15. Some questions about the calibration curves presented in Fig. 6: (1) What are the black and red lines? I guess one is MIP and the other is NIP, but it is not clear in the caption. (2) How many time the experiment was made? As the standard deviation was added, the number of experiment should be included in the caption (n = 3?). (3) Why did NIP present a linear response too? Usually, NIP synthetized by electropolymerization does not present any response in the presence of analyte, since only the polymer is formed on the electrode surface, without any cavity. Thus, the NIP profile should be the same as that obtained after MIP polymerization (before the template extraction). Because of this, it is very important to include the same results that the authors showed for MIP in Fig. 3 for the NIP as well. (I suggest that the authors check how this result for the NIP was presented in the paper Sensors and Actuators B 253 (2017) 180–186). (4) I am surprised by the wide linear range of the sensor. Sensors based on electropolymerized MIP usually has a small linear range, because the cavity is easily occupied by the analyte, resulting in saturation of the electrochemical signal. How do the authors explain this behavior? (5) With the data of the calibration curve, it is possible to calculate the affinity constant of the MIP, which can be enriching for the work.
Figure 6 was modified according to the reviewer’s suggestion.
All the experiments were made at least in triplicate, this information being now introduced in the manuscript, in the section Materials and Methods.
The NIP also presented a linear response due to the process of nonspecific adsorption of the template on the outer surface of the polymeric film. This adsorption proved to be in a logarithmic dependence with the analyte concentration in the sample, which is in accordance with the theoretical data.
Regarding the NIP profile (Figure 3 and Figure 4, which were modified to include the data for NIP), the signal of the NIP, after the extraction, was not the same as the one of the MIP after polymerization, due to the strong interactions between the polymer with numerous acidic moieties and the strongly basic solvent used for extraction. Also, the polymer obtained using I3AA as monomer, presents to a certain degree swelling properties, pH-dependent, which can help in the template removal process, but also influence the indirect response of a redox probe.
As it was already stated, the phenomenon of nonspecific adsorption heavily influenced the response, both of the MIP and NIP. These small disadvantages are caused also by the use of an indirect method of quantification, other factors being able to influence in a small degree the signal of the redox probe. But, as we also have done, these drawbacks can be surpassed by taking into account the response after extraction and after incubation and also the response for both MIP and NIP. Considering this, we chose to construct our final signal the way it is presented and also to create a pseudo-calibration curve for the NIP, in order to better represent the higher affinity of the template for the MIP.
The wide linear range can be explained by the fact that values which comprise this range are quite small and more important substantially smaller compared to the concentration used for the imprinting process, 1 µM being the upper limit of the linear range, in comparison with 500 µM used in the electropolymerization mixture. This means, that a large number of cavities per unit of surface can be formed, but due to slower kinetics a longer period being needed to reach saturation, especially at such low concentrations. Also, there are also works in the literature that present similar or even wider linear ranges for the electropolymerized MIP ([Guo et al, 1`,3`,5-Trinitrotoluene detection by a molecularly imprinted polymer sensor based on electropolymerization of a microporous-metal-organic framework, Sensors and Actuators B. 207 (2015) 960–966][Karimian et al, An ultrasensitive molecularly-imprinted human cardiac troponin sensor, Biosens. Bioelectronics, 50 (2013) 492-498. doi: 10.1016/j.bios.2013.07.013.][Mamo et al, Development of a molecularly imprinted polymer-based sensor for the electrochemical determination of triacetone triperoxide (TATP), Sensors (Basel). 14(12) (2014) 23269–23282.]).
16. It is not clear why I3AA was chosen as a monomer, since it does not form a very stable polymeric film on the electrode surface. Even with good results, what was the reason for choosing this monomer? The use of graphene or carbon nanotubes could have aided in the stability and fixation of the film on the electrode.
The I3AA was chosen as the monomer because it presents functional groups capable to interact with CFX, it can be electropolymerized in aqueous solution, at a potential that allows the use of both GCE and BDDE and its polymerization does not lead to totally insulating films, allowing the indirect detection.
The use of nanomaterials could have aided in the stability and fixation of the film on the electrode, but it would have made the electrode modification even more complicated, with more variability. We wanted to keep as a simple procedure; but the use of nanomaterials can be a further study.
17. Include the structure of all compounds used in selectivity analysis in supplementary information to facilitate the comparison and understanding among the results.
A new figure, with the structures of all the tested interferents, was added in the supplementary information.
18. How do the authors explain AMP responded to the sensor while penicillin, which as a very similar structure to AMP, did not present a signal?
We did not test penicillin, (usually, penicillin which is a class of antibiotics, not a molecule is used to designate the class; penicillin V, G, N are molecules). We only tested AMP, which belong to the penicillin class, being an aminopenicillin. The text was modified to avoid any confusion.
19. Why was the logarithmic scale concentration used in the recovery analysis (Table 1)? I could understand it in the calibration curves (since concentrations with different order of magnitude were used). However, there is no sense used logarithmic concentration in recovery analysis.
Table 1 was modified according to the reviewer’s suggestion.

Reviewer 2 Report
This manuscript describes a MIP sensor for the detection of cefalexin. A sensor for this antibiotic has been reported, but the earlier device did not involve an imprinted polymer and the authors undertook the research with the expectation that the MIP sensor would be more sensitive. This hypothesis was correct. The manuscript is sound and should be published. I have a few minor scientific questions ad a few stylistic comments that are detailed below. I believe that these items are sufficiently minor so that additional review is unwarranted.
Scientific Points
1. Why use two different electrodes? The GCE and BDDE are somewhat redundant. It is only near the end of the manuscript that the authors cite the GCE as preferred because of homogeneity, even though the BDDE was more sensitive. This was predictable and, perhaps, the authors could use that as part of their introduction to avoid readers (such as myself) from asking that question in reading the abstract.
2. In the text beginning with Line 322, the AFM measurements of roughness and thickness are relayed. How many measurements were made for each sensor? This is an important point because the MIL films are often not uniform across the film.
3. The lines in Figure 6 need to be identified in the figure caption. Which is the MIP and which is the NIP?
4. The inability to reuse the ‘sensor’ technically classifies it as a detector. This is a minor point, but should be acknowledged.
5. Line 324 notes that the sensor has low affinity for other than the template. Figure 7 shows that this is definitely not so. It has affinity for related antibiotics as well.
Stylistic Points
1. The language is mostly acceptable, but there are a number of instances in which ‘the’ or ‘a’ is missing. These should be easy to fix.
2. Line 31 should be ‘in the European’
3. Line 227 should be ‘(pH = 74) that’
4. The sentence that begins on Line 228 runs on too long and should be rewritten into several sentences for clarity.
5. The sentence that begins on Line 344 is awkwardly phrased and should be reconsidered.
6. Lines 375 to 382 should be a single paragraph not three.
7. Line 384 should read ‘probes’
8. The sentence that begins on Line 405 must be rewritten for clarity.
9. Line 411, ‘resulted’ should be ‘resulting’
10. Line 446 ‘rebounded’ should be ‘rebound’.
11. Line 461, ‘assimilated’ should be ‘simulated’
12. Line 467, ‘note’ should be ‘not’
Author Response
This manuscript describes a MIP sensor for the detection of cefalexin. A sensor for this antibiotic has been reported, but the earlier device did not involve an imprinted polymer and the authors undertook the research with the expectation that the MIP sensor would be more sensitive. This hypothesis was correct. The manuscript is sound and should be published. I have a few minor scientific questions ad a few stylistic comments that are detailed below. I believe that these items are sufficiently minor so that additional review is unwarranted.
We would like to thank to the reviewer for recognizing the quality of our study and for the useful suggestions which aim to improve the technical aspect of the work.
Scientific Points
1. Why use two different electrodes? The GCE and BDDE are somewhat redundant. It is only near the end of the manuscript that the authors cite the GCE as preferred because of homogeneity, even though the BDDE was more sensitive. This was predictable and, perhaps, the authors could use that as part of their introduction to avoid readers (such as myself) from asking that question in reading the abstract.
Both the abstract and the introduction were modified according to the reviewer’s suggestion.
2. In the text beginning with Line 322, the AFM measurements of roughness and thickness are relayed. How many measurements were made for each sensor? This is an important point because the MIL films are often not uniform across the film.
Two glassy carbon plates of 1x1 cm were modified in the same way for each type of modification (MIP, NIP, MIP after extraction). Different areas of the glassy carbon plates were analyzed and the images presented are the most representative for each type of modification; the same trend as in the case of the images presented was observed for other analyzed areas, even though, as mentioned by the reviewer, the MIP film was not entirely uniform.
3. The lines in Figure 6 need to be identified in the figure caption. Which is the MIP and which is the NIP?
The figure caption was modified.
4. The inability to reuse the ‘sensor’ technically classifies it as a detector. This is a minor point, but should be acknowledged.
Along the manuscript we used the conception of sensor as defined by IUPAC, which states: “A chemical sensor is a device that transforms chemical information, ranging from the concentration of a specific sample component to total composition analysis, into an analytically useful signal.” [IUPAC - CHEMICAL SENSORS DEFINITIONS AND CLASSIFICATION. Adam Hulanicki, Staniseaw Geab, Folke Ingman. Pure&App/. Chern., Vol. 63, No. 9, pp. 1247-1250, 1991]
5. Line 324 notes that the sensor has low affinity for other than the template. Figure 7 shows that this is definitely not so. It has affinity for related antibiotics as well.
The phrase was re-written for more clarity.
Stylistic Points
1. The language is mostly acceptable, but there are a number of instances in which ‘the’ or ‘a’ is missing. These should be easy to fix.
The entire manuscript was revised and the language level was improved.
2. Line 31 should be ‘in the European’
The manuscript was modified.
3. Line 227 should be ‘(pH = 74) that’
The manuscript was modified.
4. The sentence that begins on Line 228 runs on too long and should be rewritten into several sentences for clarity.
The manuscript was modified.
5. The sentence that begins on Line 344 is awkwardly phrased and should be reconsidered.
The manuscript was modified.
6. Lines 375 to 382 should be a single paragraph not three.
The manuscript was modified.
7. Line 384 should read ‘probes’
The manuscript was modified.
8. The sentence that begins on Line 405 must be rewritten for clarity.
The manuscript was modified.
9. Line 411, ‘resulted’ should be ‘resulting’
The manuscript was modified.
10. Line 446 ‘rebounded’ should be ‘rebound’.
The manuscript was modified.
11. Line 461, ‘assimilated’ should be ‘simulated’
The manuscript was modified.
12. Line 467, ‘note’ should be ‘not’
The manuscript was modified.

Reviewer 3 Report
The authors reported a new electrochemical sensor for the detection of cefalexin (CFX), based on the use of molecularly imprinted polymer (MIP) obtained by electropolymerization in aqueous medium of indole-3-acetic acid (I3AA). This work is sound, the detection limits are very low for the given readout principle, the measurements are reproducible, and cross-selectivity of the MIP-based sensor with respect to competitor molecules has been quantified.
I have one question for the authors. In figure curve-c, DPV voltammogram of the redox probe was observed after template removal. How can the removal of the 1-2 nm target molecule open the access of the redox marker or through the polymer layer?
Author Response
The authors reported a new electrochemical sensor for the detection of cefalexin (CFX), based on the use of molecularly imprinted polymer (MIP) obtained by electropolymerization in aqueous medium of indole-3-acetic acid (I3AA). This work is sound, the detection limits are very low for the given readout principle, the measurements are reproducible, and cross-selectivity of the MIP-based sensor with respect to competitor molecules has been quantified.
We would like to thank to the reviewer for recognizing the quality of our study and for the useful suggestions which aim to improve the technical aspect of the work.
I have one question for the authors. In figure curve-c, DPV voltammogram of the redox probe was observed after template removal. How can the removal of the 1-2 nm target molecule open the access of the redox marker or through the polymer layer?
The removal of the target molecule of 1-2 nm can be sufficient to allow the access of the redox probe to the working surface for electron transfer, considering that the redox molecule is substantially smaller. Due to the proportion of template : monomer of 1:2, during the electrogeneration of the MIP, a large number of template molecules are incorporated in the film. Thus, the removal of this large number of template molecules from the entire MIP influences the structure and the porosity of the MIP film, which influences the capacity of the redox probe to exchange electrons with the electrode surface.
